# Evaluating the diagnostic and triage performance of digital and online symptom checkers for the presentation of myocardial infarction; A retrospective cross-sectional study

William Wallace[1], Calvin Chan[1,2]*, Swathikan Chidambaram[1], Lydia Hanna[1], Amish Acharya[1,2], Elisabeth Daniels[1], Pasha Normahani[1], Rubeta N. Matin[3], Sheraz R. Markar[4], Viknesh Sounderajah[2,5], Xiaoxuan Liu[6,7], Ara Darzi[1,2]

1 Department of Surgery & Cancer, Imperial College London, St. Mary's Hospital, London, United Kingdom, 2 Institute of Global Health Innovation, Imperial College London, South Kensington Campus, London, United Kingdom, 3 Department of Dermatology, Oxford University Hospitals NHS Foundation Trust, Oxford, United Kingdom, 4 Surgical Intervention Trials Unit, Nuffield Department of Surgery, University of Oxford, United Kingdom, 5 Google Health UK, London, United Kingdom, 6 Institute of Inflammation and Ageing, University of Birmingham, Birmingham, United Kingdom, 7 University Hospitals Birmingham NHS Foundation Trust, Birmingham, United Kingdom

* calvin.chan16@imperial.ac.uk

**Data Availability Statement:** The collected anonymous confirmed myocardial infarction

## Abstract

Online symptom checkers are increasingly popular health technologies that enable patients to input their symptoms to produce diagnoses and triage advice. However, there is concern regarding the performance and safety of symptom checkers in diagnosing and triaging patients with life-threatening conditions. This retrospective cross-sectional study aimed to evaluate and compare commercially available symptom checkers for performance in diagnosing and triaging myocardial infarctions (MI). Symptoms and biodata of MI patients were inputted into 8 symptom checkers identified through a systematic search. Anonymised clinical data of 100 consecutive MI patients were collected from a tertiary coronary intervention centre between 1st January 2020 to 31st December 2020. Outcomes included (1) diagnostic sensitivity as defined by symptom checkers outputting MI as the primary diagnosis (D1), or one of the top three (D3), or top five diagnoses (D5); and (2) triage sensitivity as defined by symptom checkers outputting urgent treatment recommendations. Overall D1 sensitivity was 48±31% and varied between symptom checkers (range: 6–85%). Overall D3 and D5 sensitivity were 73±20% (34–92%) and 79±14% (63–94%), respectively. Overall triage sensitivity was 83±13% (55–91%). 24±16% of atypical cases had a correct D1 though for female atypical cases D1 sensitivity was only 10%. Atypical MI D3 and D5 sensitivity were 44±21% and 48±24% respectively and were significantly lower than typical MI cases (p<0.01). Atypical MI triage sensitivity was significantly lower than typical cases (53±20% versus 84±15%, p<0.01). Female atypical cases had significantly lower diagnostic and triage sensitivity than typical female MI cases (p<0.01).Given the severity of the pathology, the diagnostic performance of symptom checkers for correctly diagnosing an MI is

patient vignettes are available on a public repository (DOI: 10.6084/m9.figshare.25872310).

**Funding:** The author(s) received no specific funding for this work.

**Competing interests:** VS joined Google Health UK as an employee only after this work was completed. All other authors declare no financial or non-financial competing interests.

concerningly low. Moreover, there is considerable inter-symptom checker performance variation. Patients presenting with atypical symptoms were under-diagnosed and under-triaged, especially if female. This study highlights the need for improved clinical performance, equity and transparency associated with these technologies.

## Author summary

Online symptom checkers are increasingly popular tools that patients turn to in order to understand their symptoms, self-diagnose and ultimately seek further medical attention. We wanted to evaluate how accurately different commercially available symptom checkers are able to diagnose a severe medical condition (myocardial infarction) and give appropriate medical advice (i.e., seek immediate medical assistance). In this study, we collected anonymous clinical data of 100 people who had confirmed myocardial infarctions. Their presenting symptoms and biodata (e.g., age, sex, co-morbidities) were inputted into the eight most popular commercially available symptom checkers found through a Google and an App Store search. We found that the performance of the online symptom checkers for correctly diagnosing a myocardial infarction was low, especially given the severity of the disease. Additionally, there was considerable variability in performance between different symptom checkers. Finally, we found that patients with atypical presenting symptoms were less likely to be diagnosed and given correct medical advice, especially if female. Our study highlights the need for better clinical performance, equality and transparency of online symptom checkers.

## Introduction

Symptom checkers are applications or web-based tools that enable patients to input symptoms and personalised biodata to produce potential diagnoses and relevant clinical information, aiding self-diagnosis and triage. Symptom checkers are becoming an increasingly prominent feature of the modern healthcare landscape due to both increased access to internet connectivity and a cultural shift towards more involved self-care engagement. Approximately 70% of internet users search for health-related information, while over a third of adults use the internet for self-diagnosis [1–3]. Symptom checkers are especially pertinent given the unprecedented burden exerted on emergency care services, particularly in light of the COVID-19 pandemic.[3–5] There are over 23 million visits to A&E annually in the UK, of which an estimated 11.7% (approximately 2.7 million visits) of A&E attendances would be better managed by other services [6,7]. Thus, symptom checkers could reduce the financial and resource burden placed upon hospitals and can help focus resource towards those who are truly in medical need [8]. As such, governments have also incorporated symptom checkers into their formal health and social care pathways in order to alleviate the increasing burden that is placed upon both primary care services and emergency services [3–5]. In 2017, the NHS 111 triage service, backed by Babylon, reported over 15,500 app downloads and was responsible for over 9,700 triages [9]. In 2021, Babylon reportedly covered 4.3 million people worldwide, performing over 1.2 million digital consultations with 4,000 clinical consultations each day, with one patient interaction every 10 seconds.

Despite the increased use of these digital health technologies, there has been a lack of proper evaluation and assessment of public readiness which is apparent in the recent media criticism

regarding symptom checkers. The NHS-backed application Babylon was found to miss myocardial infarctions (MIs) in female users, and users with atypical presentations of chest pain [10,11]. Myocardial infarctions usually present as central, substernal chest pain that is crushing, heavy or tight in character, with possible radiation to arm, shoulder, back, epigastric, jaw or neck [12–14]. However, 20–30% of patients present atypically, especially if diabetic, elderly or female [13–16]. Delays in seeking medical attention and receiving treatment increases risk of morbidity and mortality in patients with MI [17–19]. During the COVID-19 pandemic. a 30–50% decrease in MI hospital admissions has been noted [20–22]. Concordantly during this period, the incidence of out-of-hospital cardiac arrests in England due to MI has increased by 56% [20]. This could demonstrate a reduced willingness to seek medical attention and highlights a potential use case for symptom checkers, because symptom checkers can provide urgent triage advice and enable patients to receive prompt medical attention. However, the utility of symptom checkers wholly depends on the accuracy of diagnostic and triage advice given. If a patient with a non-urgent ailment is over-triaged by the symptom checker, healthcare services may be unnecessarily utilised. Conversely, symptom-checkers that miss cases of life-threatening and serious conditions could lead to people avoiding seeking help, which may increase morbidity and mortality [8,23].

There are a number of freely available online and app-based symptom checkers that patients can utilise. However, there is currently no guidance or endorsement by health agencies such as NICE or the NHS of any particular symptom checker. Furthermore, there is often a lack of clear guidance permitting the use of symptom checkers, and where there is guidance, it is unclear how well they are followed. For example, following International Medical Device Regulators Forum (IMDRF) guidance [24], software that has a medical purpose (as defined in the guidance) qualifies as a medical device and is regulated as such. Despite this apparent international harmonisation, there is still heterogeneity between IMDRF members as to what software qualifies from jurisdiction to jurisdiction. Although there are some NICE recommended apps such as Sleepio, the majority are not registered as medical devices (Class I, II, etc under CE/UKCA/FDA), nor do they state the classification where they do qualify as a medical device. There is therefore a lack of regulatory requirements relating to safety, such as post-market surveillance requirements. Thus, the primary aim of this study is to assess and compare the performance of different, publicly available symptom checkers for diagnosing and triaging myocardial infarctions. The secondary aim is to identify the potential presence of unequitable symptom checker performance based upon symptomatology (typical vs atypical MI) and pertinent patient demographics (age, sex and ethnicity).

## Results

### Identification of symptom checkers

In total, 60 symptom checkers were identified. 11 duplicates were removed, and 39 symptom checkers did not meet the inclusion criteria. 23 of the apps were not free of charge; 9 were not in the English Language; and 3 provided a limited number of diagnoses. A further two symptom checkers were removed due to having duplicate logic tools or backend software [8]. Eight symptom checkers were finally included for the study. This included seven symptom checkers which could both diagnose and triage patients and one (SC4) that provided only diagnostic information. All the included symptom checkers had disclaimers in their terms of use that they were "not designed to provide medical diagnosis, advice, or treatment". At the time of data collection, five of the seven symptom checkers (SC1, SC2, SC3, SC6 and SC7) were registered with the MHRA by the manufacturer as Class I medical devices, the classification for the

**Table 1. Symptom checker characteristics regarding biodata, case specific questions and possible results.**

| Symptom checker | Biodata input | | History input | | | | Results | | |
|---|---|---|---|---|---|---|---|---|---|
| | Sex | Age | Medical history | Presenting complaint | Question style | Questions asked per case (mean±SD) | Number of diagnoses | MI is a possible diagnosis? | Triage advice given |
| SC1 | Y | Y | Y | Y | Y/N/IDK | 42.7 | Up to 5 | Y | Y |
| SC2 | Y | Y | Sometimes | Y | Y/N | 9.8±5.77 | n/a* | N (Serious heart problem) | Y |
| SC3 | Y | Categories | N | Y | Fill in by user | 5.2±1.05 | Variable | Y | Y |
| SC4 | N | N | N | Y | Fill in by user | 4.0±1.24 | Variable | Y | N |
| SC5 | Y | Categories | Y | Y | Fill in by user | 7.1±1.52 | Variable | N (Ischaemic heart disease) | Y |
| SC6 | Y | Y | Y | Y | Y/N/IDK | 23.0±4.75 | Variable | Y | Y |
| SC7 | Y | Y | Y | Y | Y/N/IDK | 36.9±4.79 | Variable | Y | Y |
| SC8 | Y | Y | N | Y | Fill in by user | 5.3±1.25 | Up to 5 | Y | Y |

Note

* This symptom checker could not provide alternate diagnoses once a serious heart problem was suspected.

remaining symptom checkers were unclear. Other symptom checker characteristics are summarised in Table 1.

## Participants

Of the 100 cases collected 97 were confirmed to have an acute ST-Elevation Myocardial Infarction (STEMI), whilst the remaining 3 were found to have an acute Non ST Elevation Myocardial Infarction (NSTEMI). Females made up 24% of the 100 cases collected [25]. The female group had a significantly higher mean age than males (66.8±16.6 versus 60.7±14.0, $p < .05$). The frequency of hypercholesterolaemia was also significantly higher in the female group (62.5% versus 31.6%, $p < .01$). No other significant differences were noted between groups. Demographic data is summarised in Table 2.

## Diagnostic sensitivity

The overall mean D1 (±standard deviation) sensitivity of symptom checkers was 48.0±31.4% (Fig 1). D1 sensitivity ranged from 85.0% to 6.0%. The overall mean D3 sensitivity was 72.6 ±20.2% and ranged from 92.0% to 34.0%. The overall mean D5 sensitivity was 78.5±13.5%, ranging from 94.0 to 63.0%.

**Subgroup analysis: Sex.** There were no significant differences in the mean D1 (48.7 ±33.7% versus 45.8±24.1%, $p = 0.848$) and D3 (75.7±21.7% versus 63±16.6%, $p = 0.213$) sensitivity between male and female patients across all symptom checkers (Fig 2A and 2B). For specific symptom checkers, we note that (1) D1 sensitivity for males was significantly higher than for females in SC 8 (89.5% versus 70.8%, $p < .05$), (2) D3 sensitivity for males was significantly higher than females for three of the symptom checkers assessed. Pooled mean D5 sensitivity for males was significantly higher than for females (82.1±14.5% versus 67.2±11.7%, $p < .05$) (Fig 2C). D5 sensitivity was significantly higher for males than for females in four symptom checkers examined.

**Subgroup analysis: Symptomology (atypical versus typical MI presentation).** Mean D1 sensitivity for atypical MI cases was not significantly lower than typical presentations (24.0

**Table 2. Baseline participant demographics stratified by sex.**

|  | Total | Males | Females | Significance |
|---|---|---|---|---|
| **n** | **100** | **76** | **24** | |
| **Mean age (±SD)** | 60.72 (±14.0) | 58.8 (±12.6) | 66.8 (±16.6) | **.014** |
| **Ethnicity** | | | | .830 |
| White | 30 (30) | 21 (27.6) | 9 (37.5) | |
| Black | 4 (4) | 3 (3.9) | 1 (4.2) | |
| Asian | 43 (43) | 34 (44.7) | 9 (37.5) | |
| Other / Non-Defined | 23 (23) | 18 (23.7) | 5 (20.8) | |
| **Smoking status** | | | | .266 |
| Smoker | 27 (27) | 22 (28.9) | 5 (20.8) | |
| Ex-Smoker | 11 (11) | 10 (13.2) | 1 (4.2) | |
| Non-Smoker | 62 (62) | 44 (57.9) | 18 (75) | |
| **Past medical history** | | | | |
| Diabetes | 34 (34) | 27 (35.5) | 7 (29.2) | .566 |
| Hypertension | 51 (51) | 38 (50) | 13 (54.2) | .722 |
| Hypercholesterolaemia | 39 (39) | 24 (31.6) | 15 (62.5) | **.007** |
| Myocardial Infarction | 20 (20) | 15 (19.7) | 5 (20.8) | .907 |
| **Type of MI** | | | | .459 |
| Typical | 84 (84) | 65 (85.5) | 19 (79.2) | |
| Atypical | 16 (16) | 11 (14.5) | 5 (20.8) | |

±16.2% versus 52.5±34.9%, p = .064) (Fig 3A). The D1 sensitivity for atypical MI cases was significantly lower for five symptom checkers.

Mean D3 sensitivity for atypical MI cases was significantly lower versus typical MI cases (43.8±20.6% versus 78.1±23.13%, p < .01) (Fig 3B). The D3 sensitivity for atypical MI cases was significantly lower versus typical MI cases for five symptom checkers.

Finally, mean D5 sensitivity for atypical cases was also significantly lower versus typical cases (48.4±23.6% versus 84.2±14.7%, p < .01) (Fig 3C). The D5 sensitivity for atypical MI cases was significantly lower versus typical MI cases for five symptom checkers.

**Subgroup analysis: Symptomology stratified by sex.** There were no significant differences in the mean D1 (30.7±21.7% versus 51.7±36.4%, p = .187) and D3 (54.6±24.8% versus 79.3±24.1%, p = .063) sensitivity in male patients with typical or atypical MIs (Fig 4A and 4B). One symptom checker was unable to diagnose any male atypical MI cases as a primary (D1) diagnosis. Mean D5 sensitivity for males with atypical MI was significantly lower versus males with typical MI (59.1±27.0% versus 86±15.1%, p < .05) (Fig 4C), and was evident in four symptom checkers.

Mean D1 sensitivity for females with atypical MI was significantly lower versus females with typical MI (10.0±10.69% versus 55.3±29.8%, p < .01). Four symptom checkers were unable to diagnose any female atypical MI cases as the primary diagnosis. Mean D3 sensitivity for females with atypical MI was significantly lower versus females with typical MI (20.0 ±15.1% versus 74.3±21.5%, p < .001). Mean D5 sensitivity for females with atypical MI was significantly lower versus females with typical MI (25.0±20.7% versus 78.3±15.5%, p < .001) and was seen in five symptom checkers. Two symptom checkers were unable to diagnose any female atypical MI cases in the top three or five diagnoses.

**Subgroup analysis: Age.** Stratified based on decade of birth, there were statistically significant differences within symptom checkers for D1, D3 and D5 sensitivity (p<0.05, S2 Fig). The mean D1 sensitivity was highest for users aged 50–70 years old (50.6%- 52.1%), while sensitivity was lowest for users aged 20–30 years and >90 years (37.5% and 0%, respectively). In

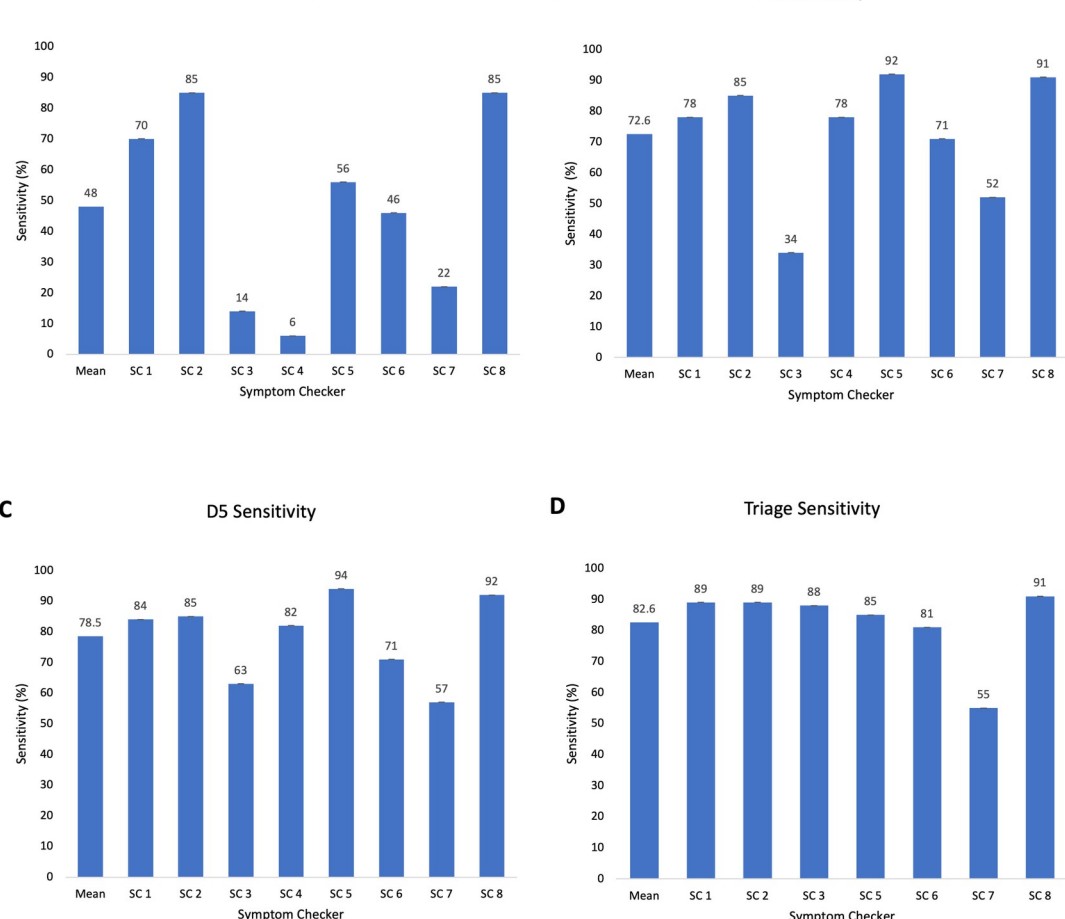

**Fig 1. Symptom checker D1, D3, D5 and Triage sensitivity for all cases (n = 100).** Note. Mean sensitivity for D1, D3, D5 and Triage has been included. SC4 was removed from Triage bar chart due a lack of triage function.

four symptom checkers, age was a statistically significant factor for D1 sensitivity with a poorer performance for ages at the extreme. The mean D3 sensitivity was highest for those aged 51–60 years (78.9%), although this decreased for older ages and reached a nadir of 0% for those aged 90 and above. At the younger extremes, D3 sensitivity was lower for those aged 30 and below (37.5%). Age was a significant factor in six of the eight symptom checkers tested for D3 sensitivity (p<0.05). The mean D5 sensitivity was highest for those aged 41–50 and 51–60 years old (82.2% and 84.4%, respectively). Most notably, the D1, D3 and D5 sensitivity was 0% across all symptom checkers for those aged 90 and above, while a similar feature was noted for those aged 30 and below in five of the eight symptom checkers.

## Triage sensitivity

The overall mean triage sensitivity was 82.6±12.6%, ranging from 91.0% to 55.0%.

**Subgroup analysis: Sex.** Mean triage sensitivity for males versus females showed no significant difference (84.6±13.0% versus 76.2±13.1%, p = .25) (Fig 2D). One symptom checker significantly higher triage sensitivity for males versus females (96.1% versus 75.0%, p < .01).

**Subgroup analysis: Symptomology (atypical versus typical MI presentation).** Mean triage sensitivity for atypical cases was significantly lower versus typical cases (52.7±20.0% versus

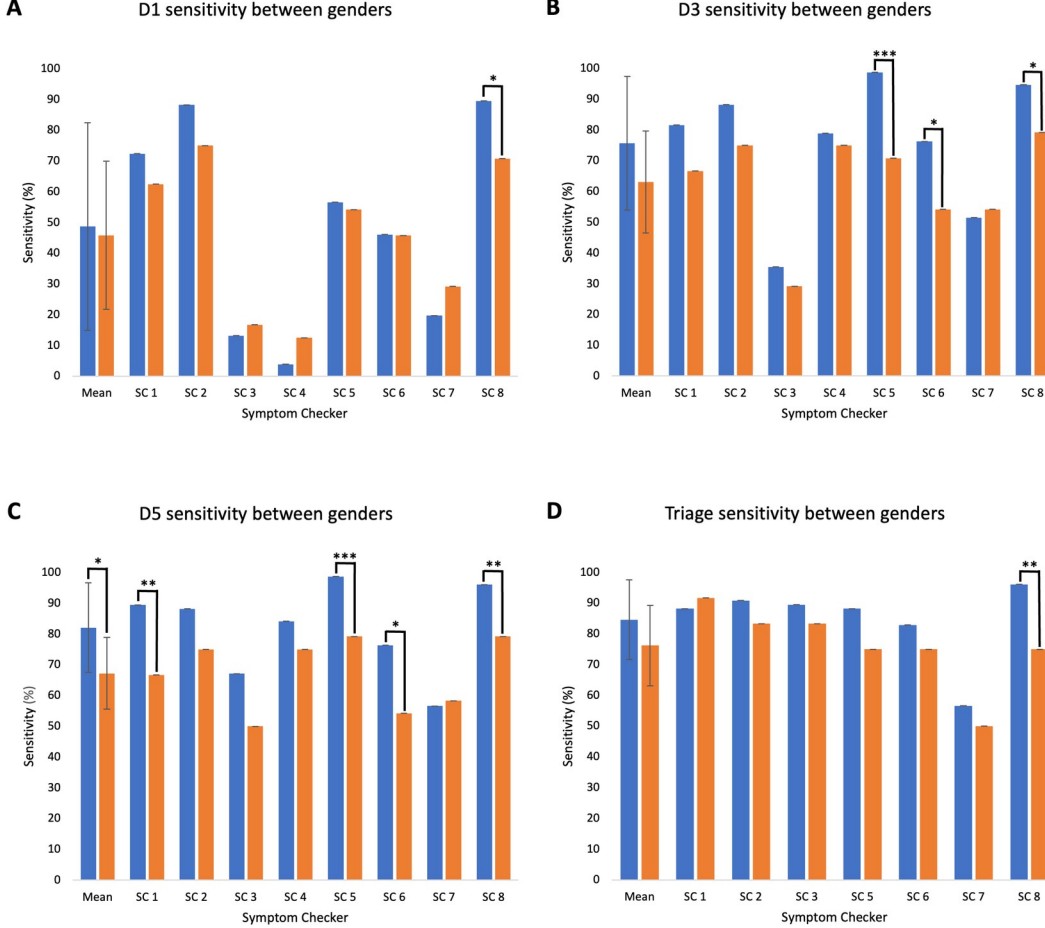

**Fig 2. Symptom checker D1, D3, D5 and Triage sensitivity for both male (blue, n = 76) and female (orange, n = 24) cases.** Note. Mean sensitivity for D1, D3, D5 and Triage has been included, error bars represent standard deviation. Overall differences between sensitivity for male and female cases assessed using t-test for each sensitivity measure. Significance within each symptom checker assessed using Pearson's Chi Squared test for each sensitivity measure. Levels of significance shown with asterisks (* p < .05, ** p < .01 and *** p < .001). SC4 was removed from Triage bar chart due a lack of triage function.

88.3±13.4, p < .01) (Fig 3D). The triage sensitivity for atypical MI cases was significantly lower versus typical MI cases for four symptom checkers.

**Subgroup analysis: Symptomology stratified by sex.** Mean triage sensitivity was significantly lower for males with atypical MI versus males with typical MI (61.0±21.5% versus 88.6 ±14.2%, p < .05) (Fig 4D). The triage sensitivity for males with atypical MI was significantly lower versus males with typical MI for five symptom checkers. Mean triage sensitivity was significantly lower for females with atypical MI versus females with typical MI (34.3±36.0% versus 87.2±10.9%, p < .01). The triage sensitivity for females with atypical MI was significantly lower versus females with typical MI for five symptom checkers. Two symptom checkers were unable to provide appropriate triage advice for any female cases with atypical MI.

## Discussion

This study evaluated the performance of symptom checkers for correctly diagnosing and appropriately triaging myocardial infarctions. The mean overall sensitivity across symptom checkers for diagnosing MI as the first diagnosis was poor and demonstrated significant

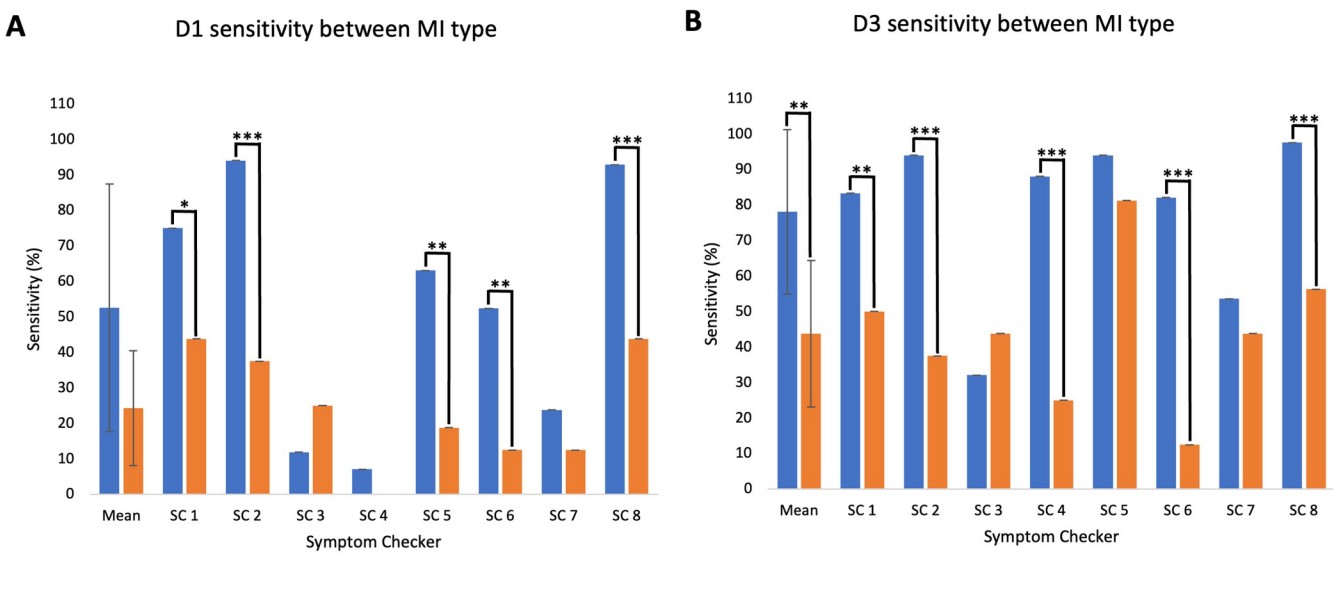

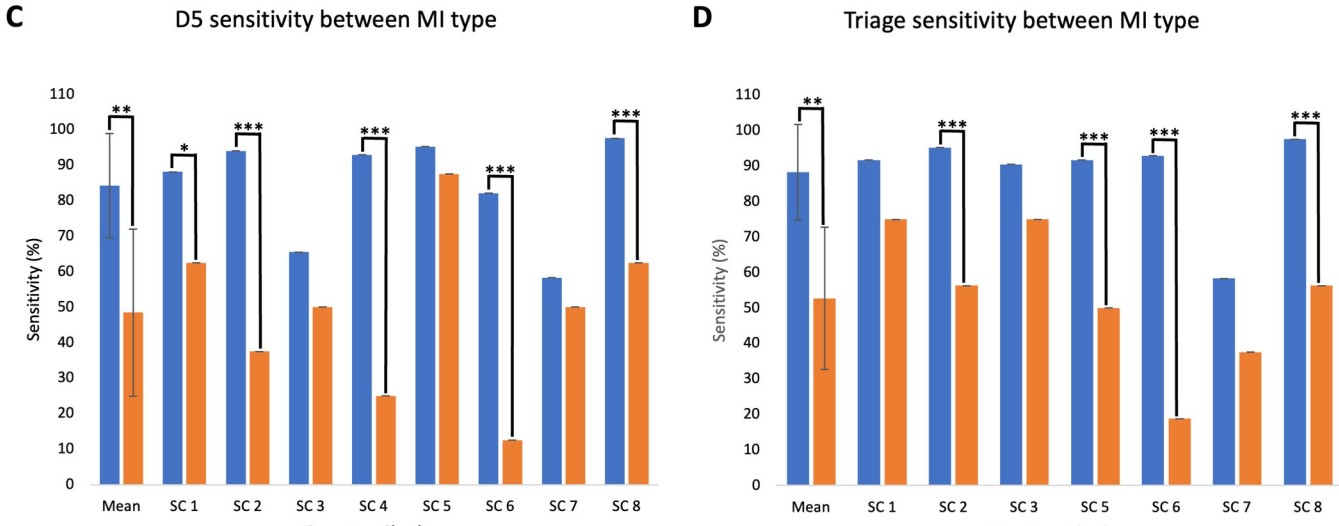

**Fig 3. Symptom checker D1, D3, D5 and Triage sensitivity for both typical (blue, n = 84) and atypical (orange, n = 16) MI cases.** Note. Mean sensitivity for D1, D3, D5 and Triage has been included, error bars represent standard deviation. Overall differences between sensitivity for typical and atypical MI cases assessed using t-Test for each sensitivity measure. Significance within each symptom checker assessed using Pearson's Chi Squared test for each sensitivity measure. Levels of significance shown with asterisks (* $p<0.05$, ** $p<0.01$ and *** $p<0.001$). SC4 was removed from Triage bar chart due a lack of triage function.

variations between symptom checkers. Additionally, variability in providing appropriate triage advice was also noted. The main factors that affected the performance of symptom checkers were an atypical pattern of presenting symptoms and sex. MI cases with atypical presenting symptoms had significantly lower triage sensitivity compared to typical MIs. Furthermore, females with atypical MI presentations were found to be significantly under-diagnosed and under-triaged than those presenting with a pattern that is more typically seen in MIs.

In this study, a primary diagnosis of MI was missed in approximately one in every two cases. This indicates a significantly low sensitivity for such a life-threatening condition. Given

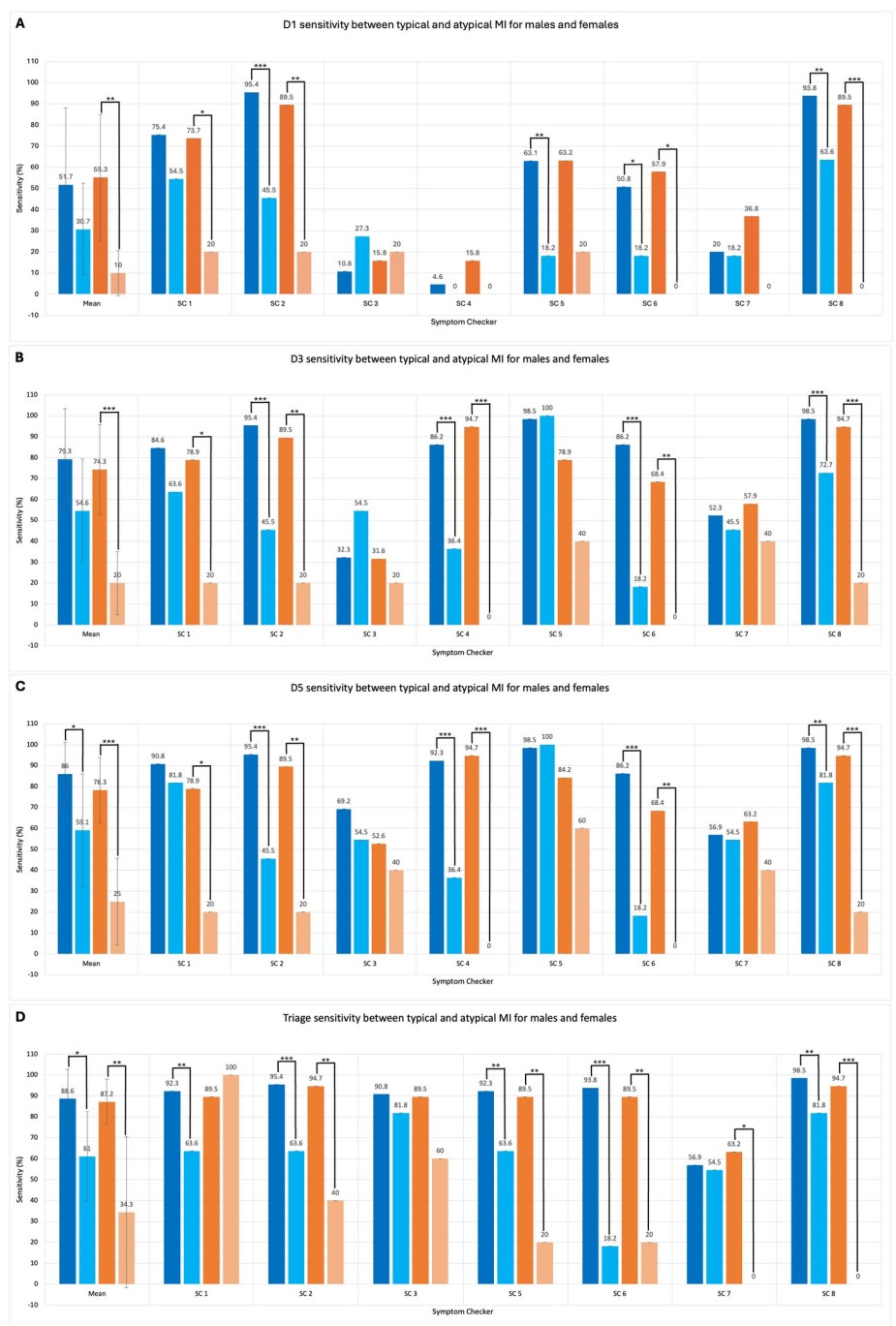

**Fig 4. Symptom checker D1, D3, D5 and Triage sensitivity for typical and atypical cases for both male (blue, n = 65 and pale blue, n = 19 respectively) and female (orange, n = 11 and pale orange, n = 5 respectively).** Note. Data labels have been included. Mean sensitivity for D1, D3, D5 and Triage has been included, error bars represent standard deviation. Overall differences between sensitivity for typical and atypical cases assessed using t-Test for each sensitivity measure. Significance within each symptom checker assessed using Pearson's Chi Squared test for each sensitivity measure. Levels of significance shown with asterisks (* $p < 0.05$, ** $p < 0.01$ and *** $p < 0.001$). One symptom checker (SC 4) was removed from Triage bar chart due a lack of triage function.

the relative lack of transparency in how these symptom checkers are constructed and validated prior to public rollout, it is difficult to pinpoint to particular factors which adversely impact performance. Several included symptom checkers claim to house "AI algorithms" as part of their diagnostic processes. AI-related factors that negatively impact performance include (1) training data that are limited (without the '4 Vs' of big data–volume, variety, velocity and veracity), (2) algorithms with limited accuracy (due to poor choice of predictive factors and poor external validity), (3) a poor user interface that prevents users from navigating the software to best utilise features, and (4) inappropriate intended use of the software. These performance factors represent potential further avenues of development for health technologists and developers (further discussed in this paper's Recommendations). However, it is currently unclear in what capacity AI is used in the decision-making process of the included symptom checkers, if at all. Indeed, symptom checkers have been reported to use hand-coded decision trees, using AI only for natural language processing to help interpret patient prompts [26,27]. Full transparency is needed in identifying the exact role of AI, if any, in these acutely patient-facing applications.

The observed variation in performance between different public-facing symptom checkers suggests a need for strict evaluation and robust regulation, given their potentially vast audience to avoid life-threatening consequences of misdiagnosis [5,23]. It is important that the evidence produced by healthcare researchers and used by regulators and policymakers is transparent and complete in order to conduct fair health technology assessments. Key factors that need to be accounted for include clinical context, reporting of reference standards and case coverage (i.e., what conditions and patient populations are accounted for). These reporting and evidential tenets are reinforced by guideline initiatives produced by The EQUATOR Network which direct the reporting of such key study characteristics for AI-centred diagnostic accuracy and clinical trial studies respectively [28,29].

The observed primary diagnostic sensitivity for detecting MI in our study was higher than previous vignette studies examining symptom checkers for other common medical conditions [8,30,31]. This could be partly attributed to the heterogenous nature of chest pain as a presenting complaint which would generate a wide list of differential diagnoses. While some of these could be MI, previous work has reported a prevalence of up to 50% of patients evaluated to have been diagnosed with non-specific chest pain, of which 56% continued to have persistent non-cardiac pain [32]. Given such a high prevalence, differentiating between imminently life-threatening cardiac pathology, less urgent non-cardiac conditions, and non-urgent non-specific chest pain is difficult without further clinical investigation. This links back to the data that can be inputted into symptom checkers. The traditional Osler's model of diagnostic process consists of history and examination before further investigations. Although symptom checkers currently do not incorporate equivalent informatic input for chest pain, the recent increase in the use of smart watches and phones which are able to collect information on vital signs such as heart rate, blood pressure and oxygen saturation is an avenue for symptom checkers to expand into and may strengthen their diagnostic and triage accuracies, although this will also rely on extremely robust data collection and require regulatory oversight.

Our study highlighted significant differences in diagnostic and triage sensitivity due to patient demographics, in particular lower accuracies for females regardless of symptomology. Symptom checkers significantly under-diagnosed females in at least one diagnostic performance measure. Although previous epidemiological studies have shown that compared to men, the overall incidence of MI is lower in women [33], it may also present with less pathognomonic symptomology [13,25]. Our findings suggest that the current algorithms utilised by the symptom checkers in our study are not sufficiently robust to discern between these presentations, which may intrinsically generate a bias against women who use these symptom

checkers. Regarding ethnicity, although differences in MI incidence depending on ethnicity has previously been noted [34,35], this study did not assess this variable. Ethnicity is likely to be a very relevant risk factor which should be considered, especially in the context of MIs where atypical presentations of chest pain are more prevalent in non-Caucasian populations [36].

Our study demonstrated that age affected the diagnostic performance of symptom checkers. Age is an important determinant in risk stratification of patients with presenting symptoms of MIs. Age is frequently incorporated in: (1) scoring the likelihood of MI as the diagnosis, (2) predicting the outcome of treatment, (3) determining patient eligibility for or benefit from interventions, and (4) in estimating the risk of short- and long-term complications. Age was a well-established risk factor for MI even before the often-cited Framingham Study. Surprisingly, in our work, the diagnostic performance was lower for ages higher than 60 in some symptom checkers. In scoring systems such as the TIMI and GRACE scores, age above 65 receives a higher score, reflecting a higher likelihood and risk of mortality and complications. This is in concordance with previous work, which has highlighted a similar age-related bias albeit in different medical conditions. Given that older patients are already predisposed to poorer health outcomes with digital technology for various reasons including a lower digital health literacy, intrinsic age-related algorithmic biases could exacerbate these outcomes [37].

Similar biases may also be present in other socioeconomic and cultural demographic factions. Particularly in light of the COVID-19 pandemic, which has already disproportionately affected disadvantaged communities, digital health technologies, including those incorporating AI, could risk exacerbating existing health inequalities [38]. A more inclusive strategy is needed so that these disparities are not further exacerbated [39]. One source of imbalance could be due to 'algorithmic bias', the application of an algorithm that exacerbates existing inequalities (e.g., socioeconomic status, ethnic background, sex and disability). Furthermore, this is compounded by questions regarding the clinical translatability and reproducibility of AI healthcare research that overall raise the issue about clinical safety [40]. Policy and decision makers, health technologists, and health officials need to be aware of these potential pitfalls.

## Recommendations

**Policy makers and regulators.** Our study emphasises an urgent reassessment of how symptom checkers are regulated that reflects the clinical function and associated risk that these systems serve in the real world as a public facing source of medical advice [23,41]. Currently, regulatory authorities, such as The Medicines and Healthcare Products Regulatory Agency (MHRA), the regulatory body responsible for medicines and medical devices in the UK, classifies Software as a Medical Device (including apps) into one of four groups (Class I, IIa, IIb and III) based on a risk classification system. The significance of the classification rules is that Class IIa and above require conformity assessment with a notified body (CE) or approved body (UKCA), whereas Class I medical devices are self-registered by their manufacture and are therefore only subject to self-certification prior to roll-out. In this study, most symptom checkers had a class 1 MHRA classification and were only subject to self-certification prior to roll-out. All of the symptom checkers had disclaimers in their terms of use that they are "not designed to provide medical diagnosis, advice, or treatment", and that patients should "discuss all information. . .with (their) physicians before making any medical decisions, including starting, stopping or modifying any medication or other treatment or care plan". This shifts the responsibility of appropriately using the apps to the lay public audience who do not necessarily have the expertise for this. Despite disclaimers, these systems clearly provide users with information that is relevant to managing disease or injury, behavioural recommendations, and can

be viewed as providing an "indicative diagnosis" in the context of lay usage. Software that provides "decisive information for making or allowing diagnosis" is sufficient for the device to be at least Class IIa under the UK Medical Device Regulations 2002 according to MHRA guidance [42]. Regulatory scrutiny should encompass the entire development pathway, from (1) the datasets used to train and validate the systems, (2) the clinical context in which evaluations were undertaken, (3) reporting of diagnostic and triage performance in the intended clinical setting of use, and (4) evaluating public understanding of the intended purpose of these systems (as this will inform how lay users utilise symptom checkers). This could all be aided by independent expert review of these systems, which will occure where the software qualifies as Class IIa and above (in the EU or UK) and where certain FDA pathways are used [43]. Finally, ensuring that symptom checkers qualify as medical devices and are appropriately classified would allow greater post roll-out scrutiny, which will supersede the current status quo of anecdotal reports through media sources.

**Health technologists.**   Robust development, validation and testing of symptom checkers is needed to account for potential biases and inequities at every stage of the health technology development process. First, development of symptom checkers that utilise an AI model in their decision-making processes should ideally incorporate real-world patient data. Moreover, larger and more diverse patient datasets should be used and algorithms should be subsequently validated on external datasets to assure its generalisability [44,45]. The importance of using real-world data has been emphasized by regulatory bodies, as seen by the introduction of the "Real-World Evidence" initiative by the FDA [46]. Second, ensuring transparent and complete reporting of datasets used is will be crucial in increasing accountability. This has been evidenced by the recent proposed "Healthsheet" initiative that set healthcare-specific standards around dataset reporting [47]. Efforts such as these to improve reporting and transparency will aid further scrutiny from regulators, policymakers, hosts of apps (app stores such as Apple and Google store) and health data institutions. Third, the roles that App stores should play in the future, including whether they should assume responsibility of validating claims as a medical device or otherwise, is yet unclear. However, App stores should actively engage in discussions about their role in the medical ecosystem, and how they can usefully contribute to protecting patient and public safety [39]. Fourth, where diagnostic dilemmas and uncertainties are common, this is not always indicated by the apps' output, so developers could be more open about this. In the same vein, integrating these applications into the care pathway means patients will be safely redirected to various components of the clinical pathway where diagnostic uncertainty is present. Finally, after deployment of the symptom checker, continuous audit should be performed. This will help find potential unequitable impacts of the software on different population groups and highlight areas of bias and other negative effects in a timely manner so that they can be addressed [39].

**Patients and the public.**   Conversely, increased patient and public awareness through education of the appropriate use and limitations of these increasingly prevalent technologies is crucial. However, patient education is hampered by the digital divide, which is driven by factors such as age, ethnicity, sex, state of health and socioeconomic status. A recent YouGov survey conducted by the British Association of Dermatologists showed that although 41% of the UK public would trust a diagnosis from smart-phone based app to diagnose skin cancer, over 50% were not confident in their ability to appraise the quality of the apps (i.e., determine whether an app can do what they claim) [48]. The inability to independently and confidently recognize and understand the quality of these apps may expose patients to using poor quality apps, which may worsen their health outcomes. Therefore, additional drivers such as health and technology literacy must also be addressed. Initiatives that have previously targeted this include NHS England's Widening Digital Participation programme, which has trained over

220,000 people, including those most vulnerable to digital exclusion, to use digital health resources [49]. Patient-driven evaluation and outcomes regarding symptom checkers are also important to better understand patient acceptability and usability of the software, and could be achieved through engaging diverse patients and public members in the development process.

## Limitations

First, the present study utilised a relatively small sample size of 100 consecutive patients from a single centre, which may limit generalisability of study findings. Additionally, power was limited for cohort subgroups (such as female atypical cases). We also found ethnic minorities to be poorly coded, with large proportions of 'unknown' ethnicities and underrepresentation of ethnic groups, such as patients. Second, data in the present study was retrospectively collected from existing routine health records. This represents a substantial limitation as data quality relies on the accuracy and completeness of clinical notes and may not wholly represent patients' first-hand symptoms. Additionally, answers given to questions asked by clinicians may differ from answers patients would have given to the symptom checker. Third, data inputted only included patients with a confirmed diagnosis of MI, which represents a selection bias. Additionally, this meant that true negatives (i.e., patients with chest pain symptomology but no MI) were not included. Fourth, two of the included symptom checkers were unable to produce a diagnosis of MI and thus their most relevant diagnosis was given, resulting in inclusion of a broader range of diagnoses such as unstable angina. This may have exaggerated the true sensitivity of these two symptom checkers. Fifth, the data collection and statistical analysis was conducted during mid-2021. Given this area of medical technology is ever-developing, it is likely that these symptoms checkers have received updates since the initial data collection. Since the data collection, the MHRA classification of the symptom checkers has or is likely to have changed and thus might not represent the current performance of these devices. Finally, performance was determined based only on cases with an established ground truth (angiography), so while we were able to evaluate the sensitivity of SCs, our analysis did not capture the specificity. Comparison against first port-of-call healthcare professionals (e.g. GP or A&E practitioner) would improve understanding of the utility of symptom checkers.

## Conclusions

This study evaluated the performance of eight symptom checkers for diagnosing and triaging myocardial infarctions. Symptom checkers generally provided low sensitivity for diagnosing MI. Triage sensitivity was higher than diagnostic sensitivty, although approximately 20% of cases were under-triaged. Patients who presented with atypical symptoms were under-diagnosed and under-triaged; especially those that were female. Our findings emphasise the need for urgent reassessment of regulation of symptom checkers from policy makers and regulators to ensure good clinical performance, whilst maintaining patient equity and technology transparency.

## Materials and methods

### Study design

This retrospective diagnostic accuracy study was conducted using anonymised "real world" patient vignettes to assess symptom checker diagnostic and triage sensitivity for confirmed cases of MI. This study has been reported in accordance with the Standards for Reporting Diagnostic Accuracy Studies (STARD) 2015 guidelines [50]. Ethical approval was granted by the Imperial College London Research Ethics Committee (reference: 22/HRA/1824).

## Participants

One hundred consecutive anonymised vignettes of patients who were treated for a confirmed MI at a tertiary primary percutaneous coronary intervention centre in London were collected between 1st January 2020 to 31st December 2020. Patients were retrospectively identified by the clinical team on the coronary care unit and cardiology ward. All consecutive adult patients who had a confirmed diagnosis of a MI were included within the time period. Ground truth was established using coronary angiography findings. Patient demographics, relevant past medical history and presenting symptomatology were extracted from electronic health records and verified by two independent investigators. Patients were coded as male or female in the electronic health system, which was assumed to be biological sex. The type of MI (typical or atypical) for each case was also noted. Atypical MI was defined by a burning chest pain or a lack of chest pain as per traditional teaching.

## Intervention

A Google internet and Apple app store search was conducted (15 March 2021) to identify symptom checkers for inclusion in this study. The first 100 results on Google were recorded, resulting in 49 potential symptom checkers being identified (S1 Fig). The top 20 most commonly downloaded results on the Apple App resulted in 11 potential symptom checkers. Eligible symptom checkers had to be free at the point of use, publicly available for use in the United Kingdom and in the English language. Symptom checkers were excluded if diagnoses were not provided; only a limited number of conditions could be diagnosed; and not in the English language.

## Test methods

Biodata (age and sex) and presenting symptoms of the 100 consecutive cases were entered into each symptom checker website or app by two independent study investigators. This simulated the results each patient would have received if they had used the symptom checker to query their own symptoms upon the onset of MI. The top five diagnoses given by each symptom checker for each case were recorded. The position of MI for each was used to measure diagnostic sensitivty, whether it being first in the list of possible conditions (D1), within the first 3 (D3) or first 5 (D5) results. If the symptom checker failed to provide a final diagnosis, this was recorded as an incorrect response. Two of the symptom checkers (SC2 and SC5) were unable to diagnose MI as a stand-alone condition; the former gave the diagnosis of a serious heart problem, while the latter diagnosed ischaemic heart disease (Table 1). Both results, respectively, were deemed correct due to this being the most serious cardiovascular condition that could be diagnosed and would include MI. Advice was considered triage in nature if the symptom checker provided an assessment of the urgency of the underlying diagnosis and/or provided advice on next steps to follow, including self-care; re-evaluation of symptoms after an interval period; seeking medical help with a GP; and attending the ED. Triage advice was deemed correct if the symptom checkers recommended contacting emergency services or the most urgent triage advice that could be produced. All other advice, such as seeking GP advice or self-care, was deemed inappropriate and recorded as incorrect. If a symptom checker reached an incorrect diagnosis but still correctly triaged the case, this was recorded as correct advice. Data collection was performed in April 2021 using the most up to date version of each symptom checker website or application and was collated on Excel (Microsoft Corporation).

## Analysis

Statistical tests were performed using SPSS Statistics 27 (IBM Corporation). All data were confirmed to be normally distributed using Kolmogorov-Smirnov tests. Two-tailed t-tests were

performed to assess for overall between-sex sensitivity differences. Pearson's Chi Squared tests were used for all other sub-group analyses. Significance was indicated if p<0.05.

## Supporting information

**S1 Fig. Flow chart showing symptom checker selection process.**
(TIFF)

**S2 Fig.** Symptom checker D1 (A), D3 (B) and D5 (C) performance for decade-wise age categories.
(TIFF)

## Acknowledgments

Infrastructure support for this research was provided by the NIHR Imperial Biomedical Research Centre.

## Author Contributions

**Conceptualization:** Rubeta N. Matin, Viknesh Sounderajah, Xiaoxuan Liu.

**Data curation:** William Wallace, Calvin Chan, Lydia Hanna, Amish Acharya, Elisabeth Daniels, Viknesh Sounderajah.

**Formal analysis:** William Wallace, Calvin Chan, Swathikan Chidambaram, Amish Acharya.

**Investigation:** William Wallace, Swathikan Chidambaram, Amish Acharya, Viknesh Sounderajah.

**Methodology:** William Wallace, Swathikan Chidambaram, Viknesh Sounderajah.

**Project administration:** Calvin Chan, Amish Acharya, Pasha Normahani, Viknesh Sounderajah.

**Resources:** Amish Acharya.

**Supervision:** Lydia Hanna, Pasha Normahani, Rubeta N. Matin, Sheraz R. Markar, Viknesh Sounderajah, Xiaoxuan Liu, Ara Darzi.

**Validation:** Calvin Chan, Lydia Hanna, Xiaoxuan Liu.

**Writing – original draft:** William Wallace, Calvin Chan, Viknesh Sounderajah, Xiaoxuan Liu.

**Writing – review & editing:** Calvin Chan, Swathikan Chidambaram, Lydia Hanna, Amish Acharya, Elisabeth Daniels, Pasha Normahani, Rubeta N. Matin, Viknesh Sounderajah, Xiaoxuan Liu.

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
