## [Decision Letter · Decision Letter 0]

5 Mar 2024

PDIG-D-23-00114

Evaluating the Diagnostic and Triage Accuracy of Digital and Online Symptom Checkers for the Presentation of Myocardial Infarction; A Retrospective Cross-Sectional Study

PLOS Digital Health

Dear Dr. Liu,

Thank you for submitting your manuscript to PLOS Digital Health. After careful consideration, we feel that it has merit but does not fully meet PLOS Digital Health's publication criteria as it currently stands. Therefore, we invite you to submit a revised version of the manuscript that addresses the points raised during the review process.

Please submit your revised manuscript within 60 days May 04 2024 11:59PM. If you will need more time than this to complete your revisions, please reply to this message or contact the journal office at digitalhealth@plos.org. Please include the following items when submitting your revised manuscript:

We look forward to receiving your revised manuscript.

Kind regards,

Daniel B. Forger

Section Editor

PLOS Digital Health

Journal Requirements:

1. Please send a completed 'Competing Interests' statement, including any COIs declared by your co-authors. If you have no competing interests to declare, please state "The authors have declared that no competing interests exist". Otherwise please declare all competing interests beginning with the statement "I have read the journal's policy and the authors of this manuscript have the following competing interests:"

2. Please provide separate figure files in .tif or .eps format only and remove any figures embedded in your manuscript file. Please also ensure that all files are under our size limit of 10MB.

3. In the online submission form, you indicated that "Supporting data may be available upon reasonable request from the corresponding author". All PLOS journals now require all data underlying the findings described in their manuscript to be freely available to other researchers, either 1. In a public repository, 2. Within the manuscript itself, or 3. Uploaded as supplementary information.

Additional Editor Comments (if provided):

Thank you for your submission to PLOS Digital Health. We apologize for the long delay in reviewing this manuscript. Finding reviewers for this particular work was more difficult than expected. However, you can see two reviews, which both point to ways that the manuscript can be significantly improved, including improvements in methodology, clarity of what was done, and in terms of citing relevant literature. We hope you will be able to address these concerns in a revised manuscript.

Reviewers' comments:

Reviewer's Responses to Questions

**Comments to the Author**

1. Does this manuscript meet PLOS Digital Health’s publication criteria? Is the manuscript technically sound, and do the data support the conclusions? The manuscript must describe methodologically and ethically rigorous research with conclusions that are appropriately drawn based on the data presented.

Reviewer #1: Yes

Reviewer #2: Partly

2. Has the statistical analysis been performed appropriately and rigorously?

Reviewer #1: I don't know

Reviewer #2: No

3. Have the authors made all data underlying the findings in their manuscript fully available (please refer to the Data Availability Statement at the start of the manuscript PDF file)?

Reviewer #1: Yes

Reviewer #2: Yes

4. Is the manuscript presented in an intelligible fashion and written in standard English?

Reviewer #1: Yes

Reviewer #2: Yes

5. Review Comments to the Author

Reviewer #1: Please see attachment as I have exceeded the character limit. I would perhaps question whether the anonymisation of the symptom checkers included meets the PLOS Data policy but I can also see why this might be necessary, the article still having value despite this limitation. I have flagged this as an issue for PLOS in the comments to the editor.

Reviewer #2: William Wallace et al. showed that the diagnostic performance of symptom checkers for correctly diagnosing an MI is concerningly low. This study highlights the need for improved clinical performance, equity, and transparency associated with these technologies. This study provides important insights for the developers and users. However, there are some concerning issues to address as follows.

1. Throughout the paper, the performance of the symptom checkers is evaluated as "accuracy". However, as the population in this study was limited to patients with MI, "sensitivity" would be a more appropriate term to use as a metric of the checkers' performance.

For the same reason, 

L224, "This indicates a significantly lower positive predictive value for such a life-threatening condition." 

L380, "so while we were able to evaluate the positive predictive value of SCs," 

"sensitivity" would probably be appropriate instead of "positive predictive value".

2. Similar to #1, in L375, "this selection bias would be expected to overestimate the sensitivity of symptom checkers for detecting MI," should be revised. Because the definition of sensitivity itself is the proportion of people who test positive among all those who actually have the disease.

3. Please give more detailed information about MI. In particular, it would be important for physician readers whether this study included only acute MI, or other types of MI, such as recent MI or OMI.

4. In the overall test of accuracy differences between the sexes, it would be better to use the Mantel-Henzel test rather than two-tailed t-tests. Please confirm the statistical analyses with a statistician.

5. Is there an explanation for the "*" in Table 1? If so, please provide the explanation.

6. PLOS authors have the option to publish the peer review history of their article (what does this mean?). If published, this will include your full peer review and any attached files.

**Do you want your identity to be public for this peer review?** For information about this choice, including consent withdrawal, please see our Privacy Policy.

Reviewer #1: Yes: Johan Ordish

Reviewer #2: No

---

## [Decision Letter · Decision Letter 1]

20 Jun 2024

Evaluating the Diagnostic and Triage Performance of Digital and Online Symptom Checkers for the Presentation of Myocardial Infarction; A Retrospective Cross-Sectional Study

PDIG-D-23-00114R1

Dear Dr Chan,

We are pleased to inform you that your manuscript 'Evaluating the Diagnostic and Triage Performance of Digital and Online Symptom Checkers for the Presentation of Myocardial Infarction; A Retrospective Cross-Sectional Study' has been provisionally accepted for publication in PLOS Digital Health.

Best regards,

Daniel B. Forger

Section Editor

PLOS Digital Health

Reviewer Comments (if any, and for reference):

Reviewer's Responses to Questions

**Comments to the Author**

1. If the authors have adequately addressed your comments raised in a previous round of review and you feel that this manuscript is now acceptable for publication, you may indicate that here to bypass the “Comments to the Author” section, enter your conflict of interest statement in the “Confidential to Editor” section, and submit your "Accept" recommendation.

Reviewer #2: All comments have been addressed

2. Does this manuscript meet PLOS Digital Health’s publication criteria? Is the manuscript technically sound, and do the data support the conclusions? The manuscript must describe methodologically and ethically rigorous research with conclusions that are appropriately drawn based on the data presented.

Reviewer #2: Yes

3. Has the statistical analysis been performed appropriately and rigorously?

Reviewer #2: Yes

4. Have the authors made all data underlying the findings in their manuscript fully available (please refer to the Data Availability Statement at the start of the manuscript PDF file)?

Reviewer #2: Yes

5. Is the manuscript presented in an intelligible fashion and written in standard English?

Reviewer #2: Yes

6. Review Comments to the Author

Reviewer #2: This is a very interesting and well-written article and the revised version is improved leading to the answers to the reviewers' comments and suggestions.

7. PLOS authors have the option to publish the peer review history of their article (what does this mean?). If published, this will include your full peer review and any attached files.

**Do you want your identity to be public for this peer review?** For information about this choice, including consent withdrawal, please see our Privacy Policy.

Reviewer #2: No
